# Restraining Quiescence Release-Related Ageing in Plant Cells: A Case Study in Carrot

**DOI:** 10.3390/cells12202465

**Published:** 2023-10-16

**Authors:** Katie Schulz, Gabriela Machaj, Paul Knox, Robert D. Hancock, Susan R. Verrall, Risto Korpinen, Pekka Saranpää, Anna Kärkönen, Barbara Karpinska, Christine H. Foyer

**Affiliations:** 1Centre for Plant Sciences, Faculty of Biological Sciences, University of Leeds, Leeds LS2 9JT, UK; katie.schulz@mac.com (K.S.); j.p.knox@leeds.ac.uk (P.K.); 2Department of Plant Biology and Biotechnology, University of Agriculture in Krakow, 31-120 Krakow, Poland; gabriela.machaj@uj.edu.pl; 3Cell and Molecular Sciences, The James Hutton Institute, Invergowrie, Dundee DD2 5DA, UK; rob.hancock@hutton.ac.uk; 4Ecological Sciences, The James Hutton Institute, Invergowrie, Dundee DD2 1BE, UK; susan.verrall@hutton.ac.uk; 5Natural Resources Institute Finland, Production Systems, Latokartanonkaari 9, 00790 Helsinki, Finland; risto.korpinen@luke.fi (R.K.); pekka.saranpaa@luke.fi (P.S.); anna.happonen@luke.fi (A.K.); 6School of Biosciences, College of Life and Environmental Sciences, University of Birmingham, Edgbaston B15 2TT, UK

**Keywords:** cell ageing, cell wall composition, lignification, metabolome, transcriptome, post-harvest processing, wounding

## Abstract

The blackening of cut carrots causes substantial economic losses to the food industry. Blackening was not observed in carrots that had been stored underground for less than a year, but the susceptibility to blackening increased with the age of the carrots that were stored underground for longer periods. Samples of black, border, and orange tissues from processed carrot batons and slices, prepared under industry standard conditions, were analyzed to identify the molecular and metabolic mechanisms underpinning processing-induced blackening. The black tissues showed substantial molecular and metabolic rewiring and large changes in the cell wall structure, with a decreased abundance of xyloglucan, pectins (homogalacturonan, rhamnogalacturonan-I, galactan and arabinan), and higher levels of lignin and other phenolic compounds when compared to orange tissues. Metabolite profiling analysis showed that there was a major shift from primary to secondary metabolism in the black tissues, which were depleted in sugars, amino acids, and tricarboxylic acid (TCA) cycle intermediates but were rich in phenolic compounds. These findings suggest that processing triggers a release from quiescence. Transcripts encoding proteins associated with secondary metabolism were less abundant in the black tissues, but there were no increases in transcripts associated with oxidative stress responses, programmed cell death, or senescence. We conclude that restraining quiescence release alters cell wall metabolism and composition, particularly regarding pectin composition, in a manner that increases susceptibility to blackening upon processing.

## 1. Introduction

Vegetables such as onions, radishes, and carrots grow underground, where it is cold and dark, especially in the winter. Carrot taproots store energy and nutrients, which are used to produce shoots that emerge from the soil in the subsequent spring when they perceive light at the surface. Globally, however, commercially grown carrots are currently stored for long periods underground in the growing fields in complete darkness to prevent light-dependent shoot growth. These industry standard procedures are applied in many countries, where unharvested carrots are stored in complete darkness in fields for over 12 months, with each field harvested as required. However, little attention has been paid to the processes that occur during the enforced ageing of vegetables such as carrots prior to processing. While senescence is a well characterized process in plants, there is a dearth of literature of the molecular physiology of ageing in plant organs, particularly those that are deprived of the environmental cues required for their further development. Recent studies demonstrate that some plant cells can have a long lifespan. For example, the parenchyma cells in Scots pine trees can be up to 42 years old, based on the number of rings containing cells with nuclei [1]. The longest-lived cells were found at the base of the stem, while the youngest were found at the base of the crown [1]. The following studies shed new light on the molecular and metabolic consequences of forcing plant organs to age by preventing access to the normal environmental cues that would trigger release from quiescence and dormancy.

The global production of carrots and turnips in 2020 was 41 million tons [2], with China producing about 44% of the world total. In many cases, harvested carrots are processed into batons of ~7 cm long, packaged and shipped to supermarkets. However, over the last 10 years, a financially crippling blackening of processed carrots has been observed. The post-harvest discoloration of fruits and vegetables is a common problem in the food production industry and a major contributor to food wastage [2,3,4,5]. However, the causes of processing-induced carrot blackening are largely unknown. Mechanical damage during harvesting, processing, or transport causes release of the plastid-localized enzyme polyphenol oxidase (PPO), which can then interact with vacuolar substrates to produce *o*-quinones, which in turn polymerize to ultimately produce high molecular weight black and brown pigments that are generically called melanin [2,3,4]. These are toxic to insects, as well as giving additional mechanical strength to tissues such as seed shells [4]. Enzymatic browning is a common cause of the discoloration seen in root vegetables [5]. In addition, the degradation of pectins in cell walls results in softening of the tissues. Pectin is present in the primary cell walls and is especially abundant in the middle lamella and cell corners. The two most common pectic polysaccharides are homogalacturonan (HG) and rhamnogalacturonan-I (RG-I) [6]. HG is a linear chain of α-1-4-linked galacturonic acid residues, whereas the backbone of RG-I consists of alternating galacturonic acid and rhamnose residues with linear and branched side chains composed mostly of arabinose and galactose residues. HG is synthesized in the Golgi in a highly methylesterified form and then secreted into the cell wall. During cell differentiation, the activity of pectin methylesterases removes methylester groups enabling HG to make a gel via calcium crosslinks [7,8]. Pectins have an important role in cell adhesion and separation, and they also regulate the cell wall pore size [7,9]. Microbial pathogens release cell wall degrading enzymes that loosen the structure of the wall [10,11]. Plants also produce these enzymes, which are essential for modifying pectin during plant growth and development [8,12]. Polygalacturonases cleave α-1-4-glycosidic bonds in de-esterified HG by hydrolysis and pectin lyases and pectate lyases by β-elimination. These pectin-degrading enzymes become increasingly active during fruit ripening. Pectin degradation also produces signals that trigger plant defense responses [13]. Cell wall rigidity is enhanced by lignin production [14]. Lignin is an important barrier against pests and pathogens as it increases the mechanical strength of the wall [15,16,17,18]. In the following studies, we have characterized the mechanisms and pathways that underpin the age-dependent blackening of cut carrots.

## 2. Materials and Methods

### 2.1. Plant Material and Sample Preparation

Carrots (*Daucus carota* variety ‘Nairobi’) were harvested from commercial fields and processed in the factories of Kettle Produce Ltd. (Balmalcolm Farm, Cupar, Fife, KY15 7TJ, UK), who provided samples of cut orange and blackened carrot batons over a three-year period. In most experiments, black and orange batons from the same harvest points were analyzed. In some experiments, samples of black, intermediate (the border between black and orange parts of the baton), and orange tissues were harvested from the same batons, as indicated in Figure 1A. 

### 2.2. Cell Wall Analyses

Sample Preparation. Orange and black carrot samples were cut by microtome into 1 cm^2^ sections and fixed in a PEM buffer (50 mM Pipes, 5 mM EGTA, 5 mM MgSO_4_, pH 6.9) containing 4% paraformaldehyde for 1 h. Fixed samples were washed twice in 1× PBS for 10 min each time and then dehydrated using increasing concentrations of ethanol (30%, 50%, 70%, 90%, and 100%) for 30 min each at 4 °C. Samples were warmed to 37 °C and incubated overnight at 37 °C in 1:1 Steedman’s wax and 100% ethanol, followed by two changes of 100% wax for 1 h at 37 °C. The samples were positioned in molds, and wax poured into the molds until a convex surface was visible, which were then set overnight at room temperature. The molds were chilled for 10 min before samples were cut into sections (Microm HM-325 microtome) and placed onto polysine-coated glass slides (VWR International, Leuven, Belgium). The sections were dewaxed and rehydrated using decreasing concentrations of ethanol (3× 97%, 90%, 50%) for 10 min each, followed by 1.5 h in water. Microscope sections were then dried.

### 2.3. Immunolabeling and Fluorescence Imaging and Processing

In this study, the following rat monoclonal antibodies (MAbs) were used: LM25, which binds to xyloglucan [19], LM5, which binds to pectic galactan [20], LM6, which binds to pectic arabinan [21], LM19, which binds to unesterified pectic HG [22], JIM7, which binds to partially methyl-esterified pectic HG [22], and LM20, which binds to highly methyl-esterified pectic HG [22].

Microscope slides containing sections of orange and black carrot tissue were incubated in 5% (*w*/*v*) milk protein in PBS for 30 min and rinsed with PBS. Monoclonal primary antibodies (1 in 5 dilution) in 5% (*w*/*v*) milk protein/PBS were applied and incubated for 90 min. Sections were washed with PBS three times for 5 min. Secondary antibodies (rabbit anti-rat IgG-fluorescein isothiocyanate (FITC) (Sigma, UK) were added (1 in 100 dilution) in 5% (*w*/*v*) milk protein/PBS and incubated for 60 min in the dark. Sections were washed with PBS three times for 5 min. To prevent interference of background autofluorescence, sections were stained with 0.1% Toluidine Blue O (pH 5.5 in 0.2 M sodium phosphate buffer) for 5 min and then the excess dye was washed off. Sections were mounted in Citifluor AF1 to decrease photobleaching. Slides were viewed with an Olympus fluorescence microscope (Olympus BX61, Southend on Sea, UK) and images captured using a Hamamatsu ORCA285 camera (Hamamatsu City, Japan) and Volocity software (Perkin Elmer, Beaconsfield, UK)

### 2.4. Measurement of Lignin

Sample Preparation. Orange and black tissues were separated, dried at 60 °C, and stored in darkness until analyzed. Tissue samples were ground to a fine powder using a Retch ball mill (Retsch MM400, Hann, Germany; 50 mL grinding jars with one metal ball, frequency 30 s^−1^). The tissue was ground in 30 s intervals 4 times to prevent heating. The extractive-free alcohol-insoluble residue (AIR) was prepared by extracting the sample 8 times with 70% (*v*/*v*) ethanol and 3 times with 100% acetone in a 25 °C sonicating water bath (30 min incubation time for each extraction with regular mixing). The ratio was 140 mg plant tissue: 14 mL solvent (70% ethanol, acetone). Samples were centrifuged (4000× *g* rpm, 10 min) before each solvent change and the pellet mixed to the solvent by vortex. The AIR was dried in a vacuum oven overnight at 60 °C. 

Lignin was quantified using an acetyl bromide (AcBr) assay [23], as modified by Kärkönen et al. [24]. Then, 5 mg of AIR and 5 mL of AcBr reagent (20% AcBr (*v*/*v*) in glacial acetic acid) were mixed and incubated at 50 °C in a heat block for 3 h with mixing by vortex every 15 min. AcBr reagent (5.0 mL) was added to an empty tube as a blank. Samples were cooled in an ice bath for 5 min, then 1.0 mL of the sample mixture was transferred to a 10-mL volumetric flask containing 2.4 mL of glacial acetic acid and 1.0 mL of 2 M NaOH. After gentle inversion, 0.1 mL of 7.5 M hydroxylamine-HCl was added and then the solution was brought to 10 mL using glacial acetic acid. A Shimadzu UV-2401 spectrophotometer (Shimadzu Corp., Kyoto, Japan) was used to measure the absorbance of the sample at 280 nm against a blank that was treated similarly as the samples. Lignin content was calculated using the following equation: Lignin% = 100(As − Ab)V/aW [As, absorbance of sample; Ab, absorbance of blank; V, volume of solution; a, absorptivity of a lignin standard (Klason lignin from spruce xylem, average from two samples 23.087 L g^–1^ cm^–1^); W, weight of sample]. Finally, 5 measurements were taken from the orange and 4 from the blackened samples.

### 2.5. Non-Cellulosic Carbohydrate Content

Non-cellulosic carbohydrate analysis was performed using acid methanolysis/gas chromatography (GC)/flame ionization detection (FID) [25]. AIR was prepared as described above. Then, 5 replicates were used for the orange samples and 4 for the blackened regions. The calibration solution contained 0.1 mg/mL of arabinose (Ara), glucose (Glc), glucuronic acid (GlcA), galactose (Gal), galacturonic acid (GalA), 4-*O*-methyl glucuronic acid (4-*O*-Me-GlcA), mannose (Man), rhamnose (Rha), and xylose (Xyl) in methanol. AIR (4 mg) prepared from ground carrots was placed in a pear-shaped, pressure resistant flask. Then, a 1 mL calibration solution was dried by evaporation and treated the same way as the carrot samples. Subsequently, 2 mL of 2 M solution of HCl in anhydrous MeOH was added and incubated for 5 h at 105 °C. Once at room temperature, the solution was neutralized with 80 µL pyridine and shaken well. An internal standard (4.0 mL) containing 0.1 mg/mL resorcinol in methanol was added and the flask shaken again. A 1-mL aliquot of the solution was evaporated using N_2_ gas. A solution containing 70 µL trimethylsilyl chloride (TMCS), 150 µL hexamethyl disilazane (HMDS), and 120 µL pyridine was used to silylate the dried sample at room temperature overnight. Samples were analyzed using GC/FID (Shimadzu GC-2010, Kyoto, Japan) with a HP-1 Column (25 m × 0.2 mm I.d., film thickness 0.11 µm). The temperature profile was as follows: 100 °C → 175 °C, 4 °C/min, 175 °C → 290 °C, 12 °C/min. The temperature of the injector was 260 °C and the temperature of the detector was 290 °C. Correction factors were used to calculate the non-cellulosic carbohydrate content; Man, Glc, and Gal 0.9, Ara and Xyl 0.88, Rha 0.89, GlcA, GalA, and 4-*O*-Me-GlcA 0.91; 2 replicates were used in all analyses.

### 2.6. Lignin Pyrolysis-GC-MS

Lignin pyrolysis-GC-MS analysis was carried out according to [26] using a Pyrola 2000 filament pulse pyrolizer with an autosampler unit. Approximately 100 μg of carrot AIR was put on a platinum filament, with a drop of acetone to keep it in place. The pyrolysis took place in a helium atmosphere at 600 °C for 2 s. The pyrolysis products were analysed using an Agilent Technologies 7890B gas chromatography system and an Agilent Technologies 5977B Single Quadrupole mass spectrometer. The pyrolysis used the following conditions: oven temperature: 50 °C, 30 s; 8 °C/min → 300 °C; 300 °C, 6 min; Column: HP-5MS 5% phenyl methyl silox, internal diameter 250 μm, length 30 m, film thickness 0.25 μm.

### 2.7. Metabolite Profiling

Gas chromatography/mass spectrometry (GC/MS) and high-performance liquid chromatography (HPLC) were performed on extracts from the orange regions, black tissues, and the border tissues that were immediately adjacent to the black regions. In addition, HPLC analysis was used to identify and quantify carotenoids. In total, 17 independent biological replicates were analysed using the GC/MS approach and 8 independent biological replicates were analyzed using the HPLC/MS and HPLC approaches. The major peaks present on the chromatograms obtained by MS were identified on the basis of parent and fragment ion masses present in the mass spectrum of each metabolite. Metabolite profiles were compared by a one-way ANOVA using carrot blackening as the single factor. A total of 64 metabolites were found to be significantly different.

### 2.8. RNA Sequencing (RNA-seq)

RNA was extracted from the orange tissues (O), black tissues (B), and the border tissues (OB/B) that were immediately adjacent to the black regions, using a CTAB method [27] followed by a RNA Clean and Concentrator^TM^ kit (Zymo Research, Irvine, California, USA) as described by the manufacturer. Three biological replicates per region were used. The quality and quantity of RNA were determined using NanoDrop ND-1000 (Thermo Fisher Scientific, Waltham, MA USA) and gel electrophoresis. Illumina-compatible sequencing libraries were prepared using the Illumina TruSeq Stranded Total RNA-with Ribo-Zero Plant kit. The libraries were checked for adaptor dimers and the insert size on a Tapestation and quantified using the Qubit system, before creating an equimolar pool of libraries. Libraries were sequenced in SE75 (single end mode, 75 bp) using NextSeq 500 (Illumina; San Diego, CA, USA) next-generation sequencing platform. RNA-Seq data was mapped to carrot reference genome ASM162521v1 ([28]; Genebank: GCF_001625215.1). We used both the reference genome and reference annotation of genes and transcripts to identify differentially expressed genes. For Gene Onthology Analysis, the list of differentially expressed genes and their functional annotation was generated using GO FEAD software (the online version is available: https://computationalbiology.ufpa.br/gofeat/index/about, accessed on 12 October 2023) by Machaj and Grzebelus [29].

### 2.9. RNA-Seq Data Analysis

Sequence data were quality checked using FastQC software followed by quality and adapters trimming in Cutadapt software. Reads trimmed to fewer than 30 nucleotides were discarded. Reads were aligned to a *Daucus carota* subsp. *sativus* reference genome ASM162521v1 ([28]; Genebank: GCF_001625215.1) using a STAR aligner [30]. The resulting alignments were checked for quality using QualiMap software [31] and Picard tools. Picard was also used to mark PCR/Optical duplicate alignments. Bioconductor R package RSubread [32] was used to extract raw counts per transcript. 

Differential expression analysis (DEA) was conducted in DeSeq2 [33] R package. The *p*-value was adjusted using the Benjamini–Hotchberg method. Differentially expressed transcripts were identified as those with an adjusted *p*-value of less than 0.05 (Appendix A). Analysis of Gene Ontology terms were made in Cytoscape with ClueGO v.2.5.7 [34] plug-in based on the functional annotation of carrot genes provided by Machaj and Grzebelus [29]. Significant (*p*_adj_ < 0.1) terms were visualized in R ggplot2 package [35]. Original data can be found in the repository Sequence Read Archive (SRA) of the National Center for Biotechnology Information (NCBI) under the project number PRJNA966197 and submission number SUB13217137.

Full details of all methods used in this study can be found in [36].

## 3. Results

### 3.1. Susceptibility to Blackening Increases with the Time of Storage Prior to Harvest

Processed carrot batons (Figure 1A,B) and transverse slices (Figure 1C,D) show a susceptibility to blackening that was greatest in the batches of carrots that had been stored underground for long periods before harvest (Figure 1E). 

Carrots harvested at or after 430 days of storage showed at least double the level of blackening after processing observed at any other age range. Carrots that had been stored for less than 161 days showed no blackening after processing (Figure 1E). Hence, the propensity to show processing-induced blackening increased with the age of the carrots.

### 3.2. Black Region Cells Are Distinct from Orange Region Cells under Bright Field Microscopy

The orange and black tissues of the batons and slices were viewed with a light microscope. Although the bright field images were taken with the same lighting and camera settings, it was more difficult to distinguish the cell walls of the orange tissues than those of the black tissues at the same settings (Figure 2A,B). The cells in the black tissues of the batons were significantly smaller than the cells in the orange regions. Discoloration, accompanied by an accumulation of dense material, was observed throughout the cells in the black tissues (Figure 2B,C). 

### 3.3. Orange and Black Tissues of Carrot Batons Have Differences in Cell Wall Polysaccharides

Monoclonal antibodies that bind to xyloglucan (LM25), HG (LM20, LM19, JIM7) or RG-I (LM5, LM6) were used to gain insight into the polysaccharide content and composition of the cell walls of black and orange tissues of carrot batons. LM26, which binds to a branched galactan epitope of RG-I, was used as a negative control antibody for the immunolabelling technique because it does not bind to carrot tissues. Cell walls of the black tissues had clearly decreased signal of xyloglucan (Figure 3f,h) compared to the orange tissues (Figure 3b,d). 

The orange and black/orange border tissues of the black batons showed strong labelling with antibodies against HG with all levels of esterification (Figure 4F,G,N,O,V,W) compared to the orange batons (Figure 4E,M,U). The black tissues, on the other hand, had greatly decreased signal of HG compared to all other tissues (Figure 4H,P,X). Interestingly, these differences in the detection of the pectic HG epitopes between orange, black, and border carrot tissues were most clear for the methyl-ester-containing epitope of JIM7 and LM20. 

LM5 and LM6 were used to detect galactan and arabinan, which are common side chains of RG-I. These side chains were detected in orange batons (Figure 5B,D), with decreased amounts in the orange and border tissues of the black carrot sections (Figure 5F,H,J,L). Cell walls in the black tissues showed the lowest detection of the LM5 and LM6 epitopes (Figure 5N,P).

Non-cellulosic cell wall carbohydrate analysis conducted by acid methanolysis/GC (Appendix A) gave similar results as immunohistochemistry. Levels of galacturonic acid, the main backbone sugar moiety in pectins, were decreased in black batons compared to orange batons. There were no significant differences in the levels of other non-cellulosic sugars in the orange and black tissues.

### 3.4. Transcriptome Reprogramming Occurs in Black Regions

Large numbers of transcripts were differentially expressed in the black and border tissues compared to the orange tissues (Figure 6). 

Significant variations in the numbers of differentially expressed genes (DEGs; Appendix A) were observed in the black and border tissues compared to the orange tissues (Appendix A). KEGG enrichment analysis of comparisons (Appendix A) revealed that transcripts encoding many metabolic processes and pathways, such as phenylpropanoid biosynthesis, TCA cycle, cysteine and glutathione metabolism, glycolysis, and gluconeogenesis, were decreased in abundance in the border and black tissues compared to the orange control tissue. However, some ontology groups, for example, cellular response to auxin stimulus, photosynthesis, and starch metabolism, were enriched in the black tissues compared to the orange control tissue (Appendix A). The levels of transcripts associated with photosynthesis, carotenoid metabolism, and starch metabolism was greatest in the border tissues (Appendix A). 

Transcripts associated with tyrosine metabolism including PPO genes (LOC108206452, LOC108192978, LOC108220626, LOC108206527) were significantly decreased in the black and border tissues (Appendix A). A large number of transcripts encoding proteins involved in the phenylpropanoid pathway were less abundant in the black and border tissues than in the orange controls. In addition, several transcripts encoding laccases and peroxidases, as well as reticuline oxidases, were less abundant in black tissues (Figure 7). Interestingly, a gene encoding a polygalacturonase, which is the major enzyme responsible for pectin disassembly during fruit ripening, was highly induced in the black regions.

A number of transcripts increased in the black tissues compared to the orange controls encode components of phytohormone signaling pathways (Appendix A). For example, auxin-responsive protein IAA26 was in the list of transcripts that were most significantly increased in the black tissues compared to the orange controls (Figure 7, Appendix A). In addition, transcripts associated with strigolactone and ethylene signaling were also highly expressed in the black tissues (Appendix A). Only one transcript associated with hormone functions appears in the list of transcripts that were most decreased in the black tissues compared to the orange controls (Figure 7). This encodes ethylene-responsive transcription factor ERF113, which is an activator of plant development and stress tolerance pathways that functions downstream of ABA signaling. 

The levels of large numbers of transcripts encoding transcription factors were changed in the black and border tissues relative to the orange tissues (Appendix A), including members of the WRKY, bHLH (B), ERF, TCP, MYB, and GATA families. The levels of NRT1/PTR FAMILY 2.11-like transcripts, which encode a high-affinity, proton-dependent glucosinolate-specific transporter, were also increased. NRT1 is involved in the apoplastic phloem-loading of glucosinolates. 

A number of transcripts were increased in the border tissues between the orange and black regions compared to orange tissues (Figure 8). Several of these, such as PHYTOCHROME KINASE SUBSTRATE 1, which is a phototropin 1-binding protein required for phototropism, are involved in the control of growth. Similarly, levels of transcripts encoding a gibberellin-regulated protein 14 were increased in the border regions. This transcript encodes a GASA domain containing protein, which regulates increases in plant growth through GA-induced and DELLA-dependent signal transduction. The most highly downregulated transcripts in the border tissues encode components associated with cell wall metabolism (Figure 8).

Of the transcripts that were increased in the black tissues compared to border tissues (Figure 9), HIP1 encodes an E3 ubiquitin-protein ligase that mediates ubiquitination and subsequent proteasomal degradation of target proteins, and NDR1/HIN1-like protein 12 plays a crucial role in plant responses to biotic stress. In addition, wound-induced protein 1-like and dormancy-associated protein 1-like were highly expressed in the black tissues compared to border tissues (Figure 9), as was GLABRA2 Expression Modulator (GEM)-like protein 8 that encodes the GL2-expression modulator, which is involved in the spatial control of cell division, patterning and differentiation in root epidermal cells, and ethylene responsive 8, which is involved in both ABA and immune signaling. Out of cell wall modifying enzymes, a gene encoding pectinesterase 2 was induced in the black tissues compared to the border tissues (Figure 9). The list of transcripts with significantly lower abundance in the black tissues compared to border tissues (Figure 9), includes auxin-responsive protein SAUR71-like, which is an early auxin response gene, and the bHLH transcription factor bHLH30-like and ERF113, which is involved in abiotic stress responses.

### 3.5. The Metabolic Profiles Are Significantly Changed in Orange and Black Carrot Regions

In total, 94 metabolites were identified using HPLC/MS, HPLC, and GC/MS analysis of black, border, and orange carrot segments. Figure 10 provides an overview of the changes in primary metabolites observed in the black tissues compared to orange regions. The black tissues showed a general decrease in primary metabolites, particularly sugars and amino acids. Conversely, increases in the abundance of lysine and γ-aminobutyric acid (GABA) were observed in the black samples. A large decrease in tricarboxylic acid (TCA) cycle intermediates, including fumarate, malate, and citrate, was observed in the black tissues (Figure 10). Of the 19 amino acids identified in the black and orange carrot samples using GC/MS, the levels of 16 were significantly decreased in the black compared to the orange samples. 

The levels of several amines/polyamines were significantly changed in the orange and black carrot samples (Appendix A). In particular, the levels of allantoin and putrescine were increased in the black samples. A decrease in the levels of soluble fructose, glucose, sucrose, and inositol was observed in the black carrot samples but the levels of mannose, galactose, glycerol, and mannitol were increased compared to the orange tissues (Appendix A). In particular, mannitol was 10 times more abundant in the black than the orange samples. The levels of threonic and galactaric acids were increased in the black samples. Significant increases in the levels of most of the 19 identified fatty acids were observed in the black carrot segments but no significant differences in the major carrot carotenoids were found (Appendix A). However, there was a significant decrease in lutein levels in the black compared to the orange carrot regions. 

There was a general increase in the levels of soluble phenolic compounds in the black carrot segments (Figure 11). In particular, significant increases in the levels of chlorogenic acid, caffeic acid, dicaffeoylquinic acid, and 5-caffeoylquinic acid were found. 

Interestingly, phenylalanine levels were lower in the black tissues, while many intermediates in the phenylpropanoid pathway and products of the pathway were increased. Moreover, the lignin content of the black tissues was more than double that of the orange tissues (Figure 11). Differences in lignin subunit composition were observed between the orange tissues and the black regions, with S units only present in black tissues in addition to more abundant G and H units (Figure 12). 

## 4. Discussion

Plants synchronize their life history according to seasonal cues and triggers. Little attention has been paid to the fitness consequences if plant organs are deprived of appropriate environmental cues that trigger release from quiescence and/or dormancy, or the associated effects of ageing caused by preventing completion of the natural life cycle. Plant organs can delay senescence until after they reproduce successfully, the timing of senescence being influenced more by developmental age than calendar age [37]. Carrots have a two-year life cycle, with vegetative growth and formation of a large tap root in the first year. The carrots enter endodormancy and they are at their peak for harvest. If unharvested in the first year, cold-induced ectodormancy can occur in the tap roots over winter. Carrots are thus harvested in the dormant state. The mechanisms of quiescence/dormancy release in carrots and the effects of restraining this release are poorly documented in the literature. However, the decreases in starch and sugars observed in the black tissues of cut carrots suggest that processing triggers release from the quiescent/dormant state. 

The data presented here reveal the complexity of transcriptome and metabolome reprogramming that occurs when aged carrots are processed. The slow processes that lead to the deposition of black deposits are only observed in aged carrots that have been stored underground for 12 months or over. Carrots that are over 1 year and 2 months old are highly susceptible to wound-induced blackening at values twice as high as in carrots in younger age ranges. These findings support the conclusions that restraining dormancy release in carrot roots is the major factor causing susceptibility to blackening. The appearance of black tissues in processed carrots followed a similar timescale in carrots harvested from all field sites, with the highest level of blackening occurring either on the day of processing or the following day [36]. The process of wounding alone does not cause the blackening because color changes do not occur in carrots that are under 12 months of age. Hence, the activation of the blackening response to processing is age dependent. 

Substantial reprogramming of gene expression and metabolism occurred following the processing of aged carrots. This incorporates a shift to secondary metabolism, with an accumulation of phenolic compounds (Figure 11), particularly chlorogenic acid and lignin with syringol (S) units (Figure 12) in the black regions, together with more abundant guaiacol (G) and *p*-hydroxyphenyl (H) units. Earlier studies on carrot taproots have also reported the predominance of G units in lignin, with increase in lignin-coupled *p*-hydroxybenzoate with maturation [38]. However, transcripts encoding phenylpropanoid pathway components were lower in the black regions. Transcripts encoding enzymes involved in phenylalanine metabolism, glutathione metabolism, and the fatty acid degradation pathways were all lower in black than orange tissues. Since the black cells remain viable, these findings are somewhat surprising given that phenolic compounds associated with PPO, including chlorogenic acid, caffeic acid, and 5-caffeoylquinic acid, were enriched in the black regions. The transcriptome data clearly indicate that the expression of genes involved in secondary metabolism is constrained following the accumulation of secondary metabolites. This regulation may serve to prevent starvation resulting from the depletion of primary metabolites in the black carrot regions. Amino acids, soluble sugars (glucose, fructose, sucrose), and organic acids are depleted in the black compared to the orange regions. This may be related to the increases in respiration following wounding. The levels of most of the TCA cycle organic acids were decreased in the black regions, indicating a depletion of respiratory substrates. The low levels of glucose, fructose, and sucrose also suggest that the black tissues were running out of vital carbohydrate reserves. 

The observed decreases in the aromatic amino acids support the conclusion that the blackening process involves a switch from primary to secondary metabolism. The dramatic ~8-fold increase in GABA in the black tissues is indicative of an abiotic stress response. A strong correlation between succinic acid and GABA was observed in carbon starvation-induced GABA production in Arabidopsis leaves [39]. Jasmonate (JA) synthesis was increased in wounded lettuce leaves, together with amplified wound signaling through the oxylipin pathway associated with leaf browning [40]. Moreover, JA plays a role in dormancy release in cold stratified wheat grains through suppression of ABA-synthesis genes [41]. While decreased levels of transcripts involved in ABA signaling were observed in the black and border regions, the levels of only one transcript associated with JA signaling was increased, and there was no evidence of changes in the expression of genes associated with JA synthesis. 

The expression of genes encoding proteins involved in auxin signaling, as well as ethylene-responsive transcription factors, was increased in the black tissues compared to the orange regions. In contrast, DEGs encoding key enzymes involved in SA synthesis, such as chorismate synthase, aminodeoxychorismate synthase, and several chorismate mutases, were decreased in the black and border regions. Other genes encoding proteins associated with programmed cell death (PCD), such as the lesion simulating disease (LSD)-like proteins and accelerated cell death 6-like proteins, were also decreased in abundance in black and border regions. However, two LSD-like and three accelerated cell death proteins, which are negative regulators of PCD were more abundant in the black and border regions. 

As with other phytohormones, auxin plays a key role in bud dormancy in carrot taproots. The endogenous levels of auxin, ethylene, cytokinin, ABA, and gibberellic acid (GA_3_) increase in the taproot up to harvest time and subsequently decrease upon harvest [42,43]. The changes in hormone levels facilitates a short dormancy period after harvest because of preharvest hormonal accumulation. The levels of cytokinin and auxin increase when the dormancy period ends, while ABA, ethylene, and GA_3_ levels decrease. The transcriptome data reported here suggest that auxin levels are high in the black and border tissues compared to orange carrots. While auxin signaling is linked to lignification through the induction of ethylene biosynthesis [43], there is no evidence to suggest that changes in auxin levels also increase susceptibility to blackening following processing. However, auxin plays a key role in regulating the degree of pectin and HG methylesterification. The data presented here suggest that blackening is related to cell wall modifications, which are linked to secondary metabolic pathways and to intercellular communication [44]. Pectin is a major contributor to signaling, with roles in stem cell maintenance, cell elongation and morphogenesis, and cell to cell signaling [44,45]. Data presented here show that blackening was accompanied by large changes in the cell wall structure, with a decreased abundance of pectins (homogalacturonan, rhamnogalacturonan-I, galactan, and arabinan) and xyloglucan. These findings demonstrate that cell wall degradation was enhanced in the black regions. Aspects of the observed cell wall modulations between the orange, border, and black carrot tissues may also relate to changes in the cell wall structure that in turn influence the access of the antibody probes. For example, the observed autofluorescence of cell walls of black regions, likely reflecting extensive cross-linking of phenolic compounds, may prevent polysaccharide detection. Substantial changes to the cell walls occurred in the black regions. The levels of galacturonic acid, which is the main component of pectin, were lower in the black tissues. The cell walls in the orange tissue were rich with HG with low, medium, and high levels of esterification. There was also a strong signal of the RG-I sidechains (1-4)-β-D-galactan and (1-5)-α-L-arabinan in orange tissues, with much lower levels detected in the border tissues and little to no detection in the black regions. These findings may suggest that pectin is one of the first compounds to be altered or removed when the blackening process is triggered. Pectin-degrading enzymes are commonly secreted by fungal or bacterial plant pathogens to weaken cell walls [10,11,41], but they are also produced endogenously by plants, for example during fruit ripening [8]. Pectic fragments, which are released by pectic enzymes following pathogen attack, are detected by the host as danger signals, called damage-associated molecular patterns (DAMPs), which reprogram the plant transcriptome and metabolome. Similarly, oligogalacturonides (OGs), the breakage products of HG, are a major source of cell wall-derived DAMPs [46,47]. 

Unlike the situation in animals, there is a dearth of literature on the ageing process in plant cells. More work is required to understand the plant cell ageing process and how it affects the cell biology and hence stress responses of developmental organs, when quiescence/dormancy release is prevented. The data presented here suggest that the answer resides at least in part in changes in cell wall metabolism that is accelerated by processing leading to molecular and metabolic reprogramming.

## 5. Conclusions

This manuscript describes a new phenomenon, i.e., the age-dependent, processing-induced blackening of carrots that have been artificially aged underground. These results shed new light on the cellular, molecular, and metabolic processes that contribute to the blackening of plant organs, in which the release from quiescence/dormancy has been restrained. The data presented here strongly implicate cell wall-related changes and associated signaling in susceptibility to blackening. The massive shift from primary to secondary metabolism observed in the black tissues, resulting in lignification and cell wall polysaccharide modification, may benefit organs that are waiting to receive the environmental cues required to trigger their further development. The age of the carrots at the time of processing is clearly a major factor controlling the susceptibility to blackening. From an industry perspective, the easiest way to avoid blackening of processed carrots would be to stop processing carrots that had been stored underground for over 12 months. In addition, nitric oxide is an important regulator of dormancy in seeds and buds that maintains the postharvest quality of fruits and vegetables during storage through the regulation of gene expression and the post-transcriptional modification of proteins, such as tyrosine nitration, S-nitrosylation, and nitroalkylation [48]. It may be that the application of a NO donor to processed carrot batons may be sufficient to prevent the processing-induced blackening of aged carrots. 

## Figures and Tables

**Figure 1 cells-12-02465-f001:**
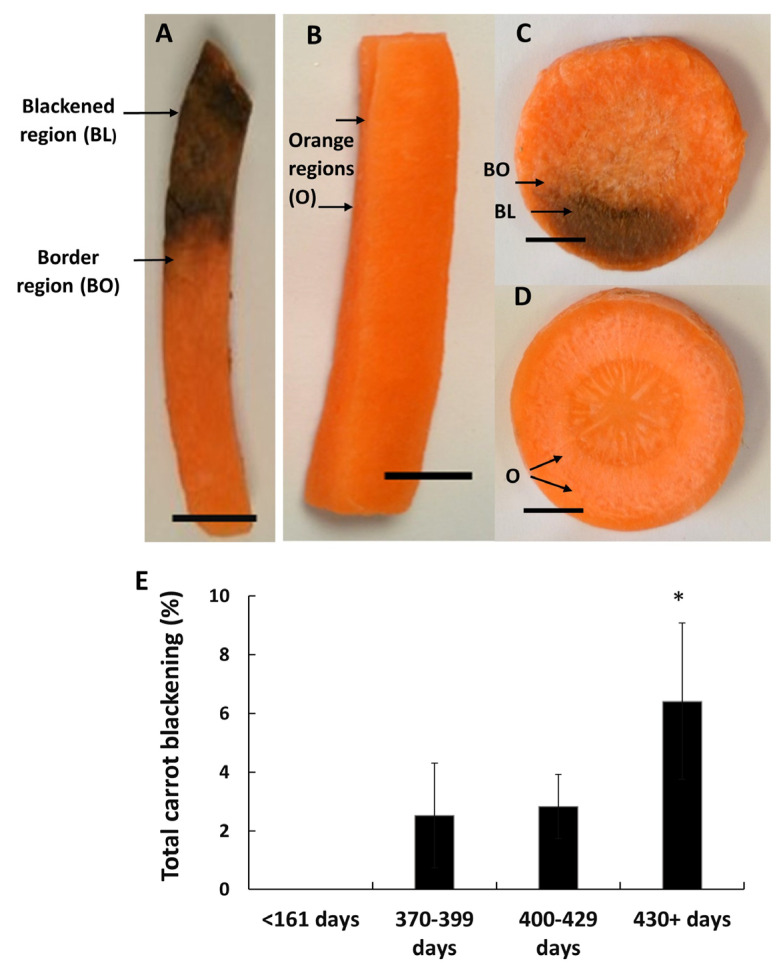
Examples of the orange and blackened carrot batons and slices, together with the effect of carrot age on the percentage spoilage of processed carrots. Samples were prepared from black (**A**) and orange (**B**) batons prepared from 400-day old carrots. Transverse slices were prepared from black (**C**) and orange (**D**) 400-day-old carrots. Labelling indicates the points on the samples where tissues were harvested for analysis. Samples were prepared for analysis from the orange tissues of non-blackened batons and slices, as well as the orange tissues from blackened batons and slices, together with blackened and border tissues. The effect of carrot age at the time of harvest on the percentage spoilage of processed products (**E**). Significant differences were calculated by student *t*-test. * *p* ≤ 0.05 and error bars represent ± SE. Scale bar = 1 cm.

**Figure 2 cells-12-02465-f002:**
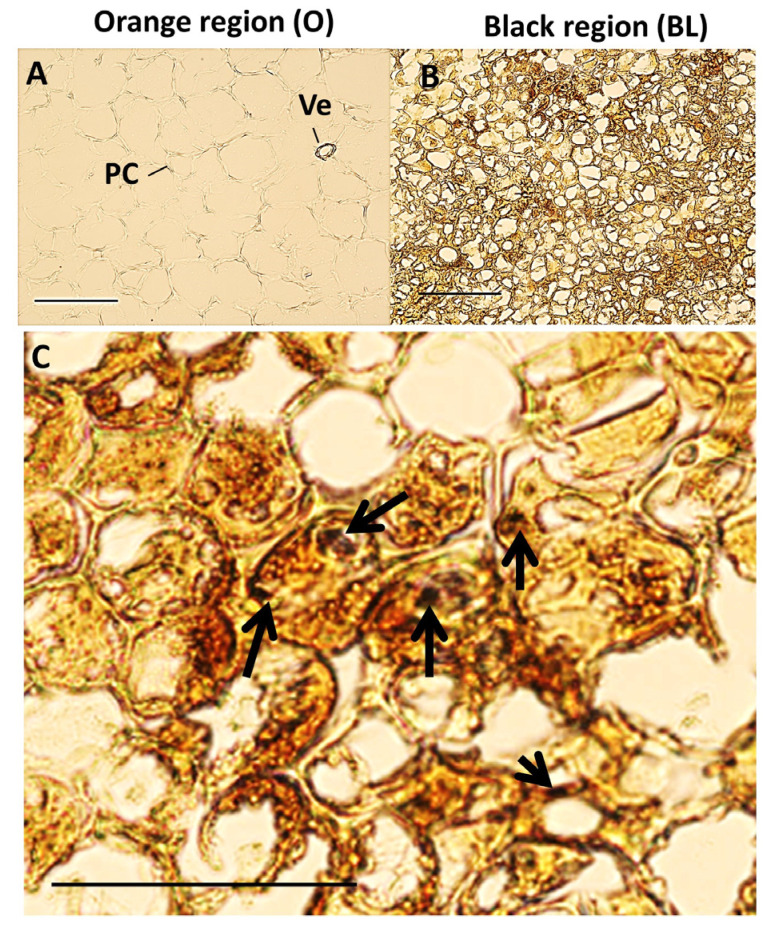
Micrographs of orange and black carrot regions. Images were taken from 12 μm thick unstained microtome sections by using the same microscope settings; orange carrot tissue (**A**) and black carrot tissues (**B**,**C**). Arrows indicate areas of strong blackening. Scale bars = 10 μm and 20 μm. PC: parenchyma cells; Ve: Vein.

**Figure 3 cells-12-02465-f003:**
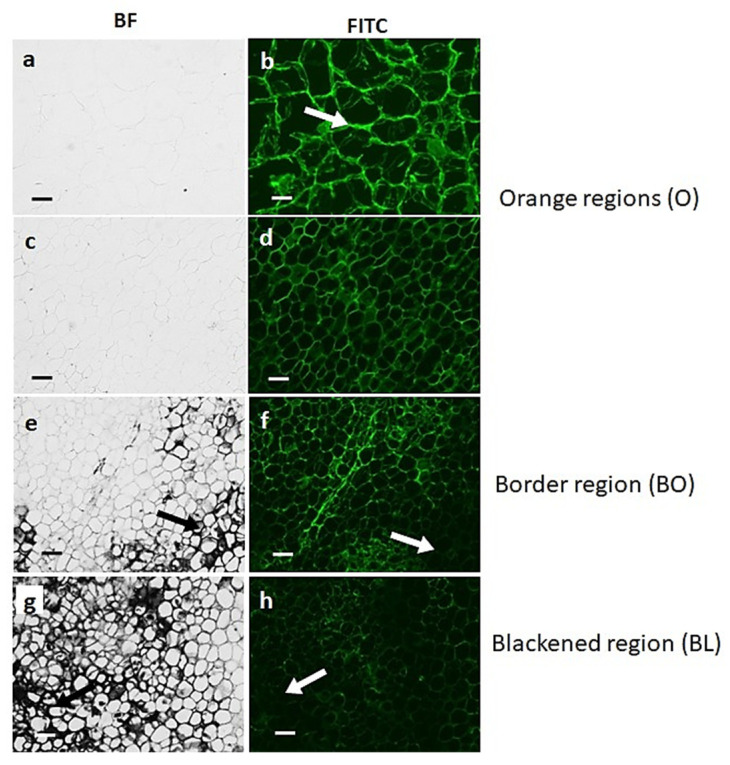
Indirect immunofluorescence detection of xyloglucan with LM25 antibody in transverse sections of carrot batons. Brightfield (BF) images showing different carrot sections: orange regions O (**a**–**d**), border tissue directly bordering the black region BO (**e**,**f**), and the black region BL (**g**,**h**). Corresponding immunofluorescence images (FITC) for xyloglucan. Scale bar = 10 μm. Exposure times: BF = 0.045 s, FITC = 0.15 s. Arrows indicate areas of strong blackening.

**Figure 4 cells-12-02465-f004:**
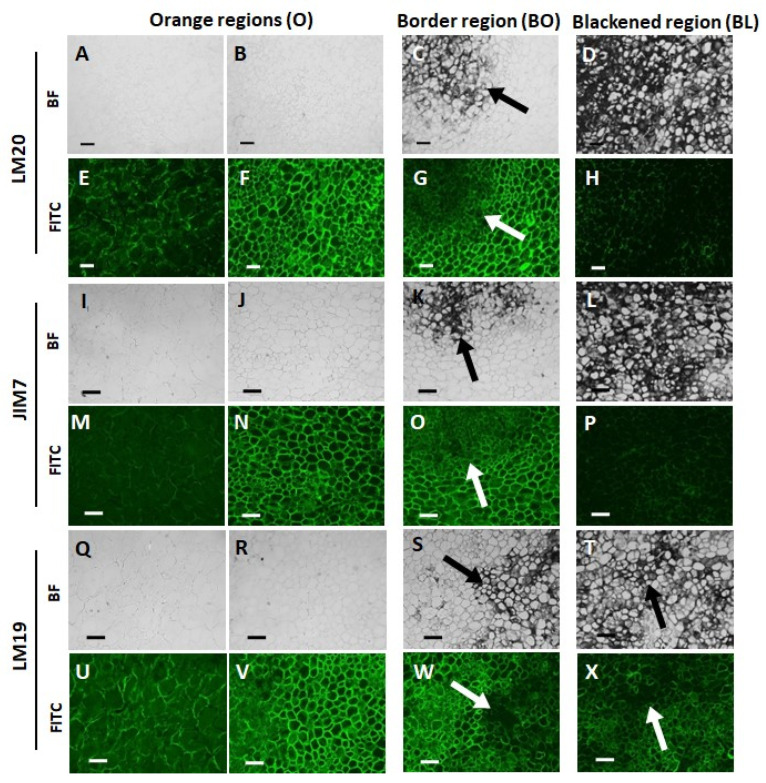
Indirect immunofluorescence detection of pectic homogalacturonan (HG) in transverse sections of orange and blackened carrot batons. Bright field (BF) images showing orange regions (O) (**A**,**B**,**E**,**F**,**I**,**J**,**M**,**N**,**Q**,**R**,**U**,**V**), tissue directly bordering the black region BO (**C**,**G**,**K**,**O**,**S**,**W**), and the black region BL (**D**,**H**,**L**,**P**,**T**,**X**). Corresponding immunofluorescence images taken in the fluorescence (FITC) channel generated using monoclonal antibodies against de-esterified pectin (LM19), methyl-esterified pectin (JIM7) and highly methyl-esterified pectin (LM20). Scale bar = 10 μm. LM20 and JIM7 exposure times: BF = 0.011 s, FITC = 0.15 s. LM19 exposure times: BF = 0.011 s, FITC = 0.2 s. Arrows indicate areas of strong blackening.

**Figure 5 cells-12-02465-f005:**
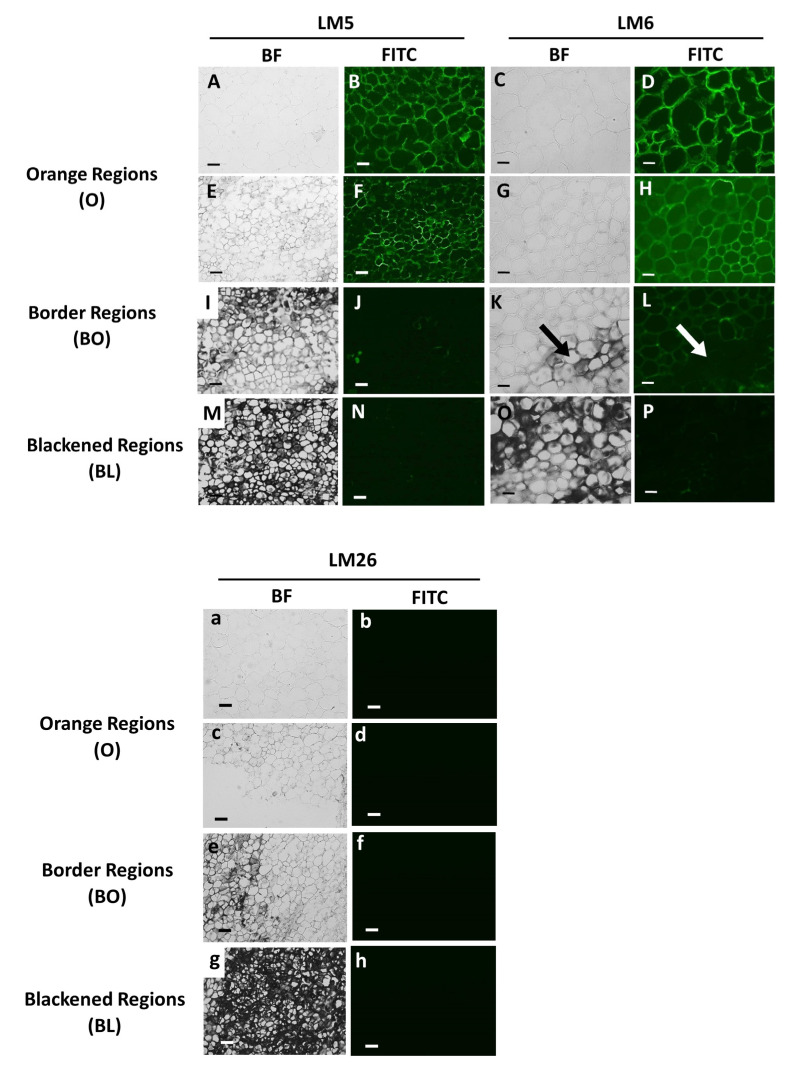
Indirect immunofluorescence detection of pectic RG-I in transverse sections of orange and blackened carrot batons. Bright field (BF) images showing tissue of an orange regions (**A**,**C**,**E**,**G**,**a**,**c**,), tissue directly bordering the black region BO (**I**,**K**,**e**,) and the black region BL (**M**,**O**,**g**) Corresponding immunofluorescence images taken in the fluorescence (FITC) channel after labelling with monoclonal antibodies after binding to (1-4)-β-galactan (LM5), (1-5)-α-L-arabinan (LM6) and branched (1-4)-β-galactan (LM26). FITC images of orange regions (**B**,**D**,**F**,**H**,**b**,**d**), tissue directly bordering the black region BO (**J**,**L**,**f**) and the black region BL (**N**,**P**,**h**). Scale bar = 10 μm. LM5 images are at 10× magnification. Exposure times: BF = 0.045 s, FITC = 0.3 s. LM6 images are at 20× magnification. Exposure times: BF = 0.1 s, FITC = 0.8 s. Arrows indicate areas of strong blackening.

**Figure 6 cells-12-02465-f006:**
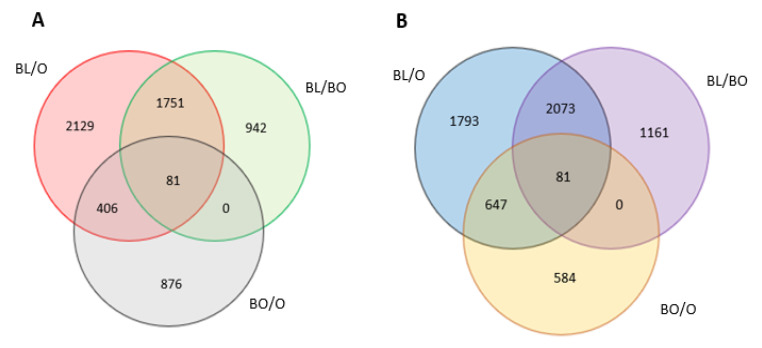
Venn diagrams showing transcript increased (**A**) and decreased (**B**). Black vs. orange (BL/O), border vs. orange (BO/O) and black vs. border (BL/BO) regions. *p*-values were calculated using Deseq2 tool and later adjusted Benjamin-Hochberg correction (*p* < 0.05; fold change ≥ 0.2).

**Figure 7 cells-12-02465-f007:**
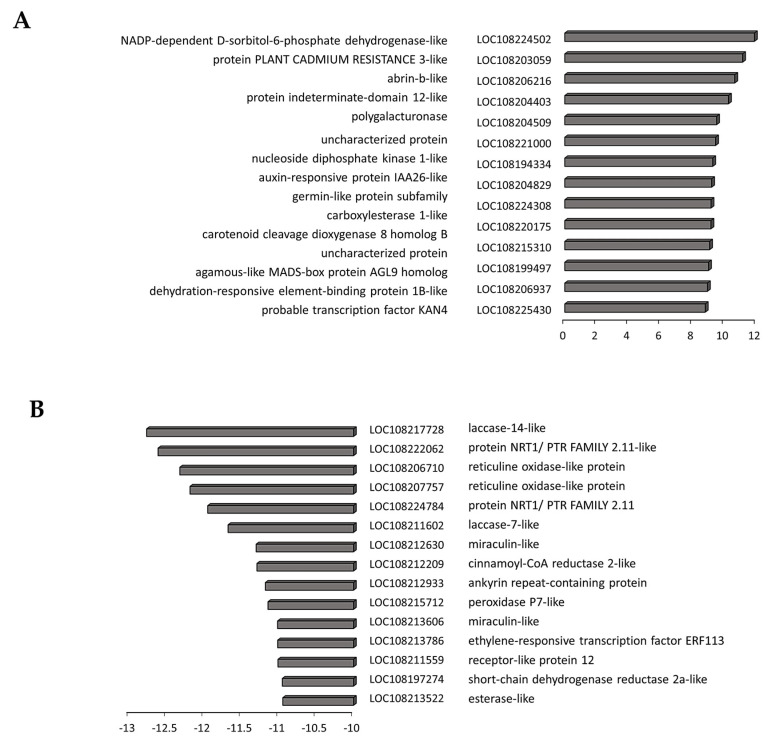
The 15 most increased (**A**) and decreased (**B**) transcripts in black tissues compared to orange tissues. Three biological replicates. *p*-values were calculated using Deseq2 tool and later adjusted using the Benjamini-Hochberg procedure (*p* < 0.05; fold change ≥ 1). Values represent log2(Fold change).

**Figure 8 cells-12-02465-f008:**
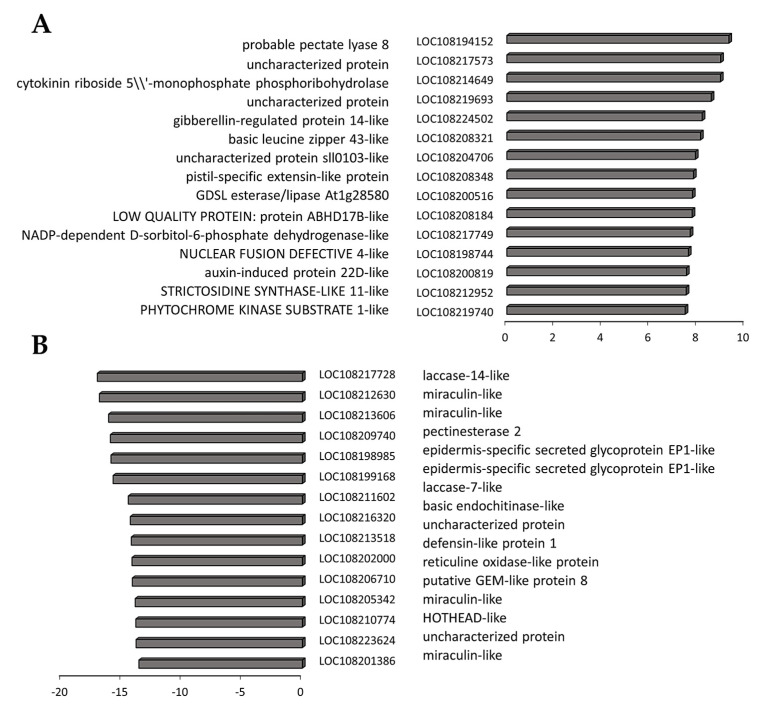
The top 15 most increased (**A**) and decreased (**B**) transcripts in the border tissues compared to orange tissues. Three biological replicates. *p*-values were calculated using Deseq2 tool and later adjusted using the Benjamini-Hochberg procedure (*p* < 0.05; fold change ≥ 1). Values represent log2(Fold change).

**Figure 9 cells-12-02465-f009:**
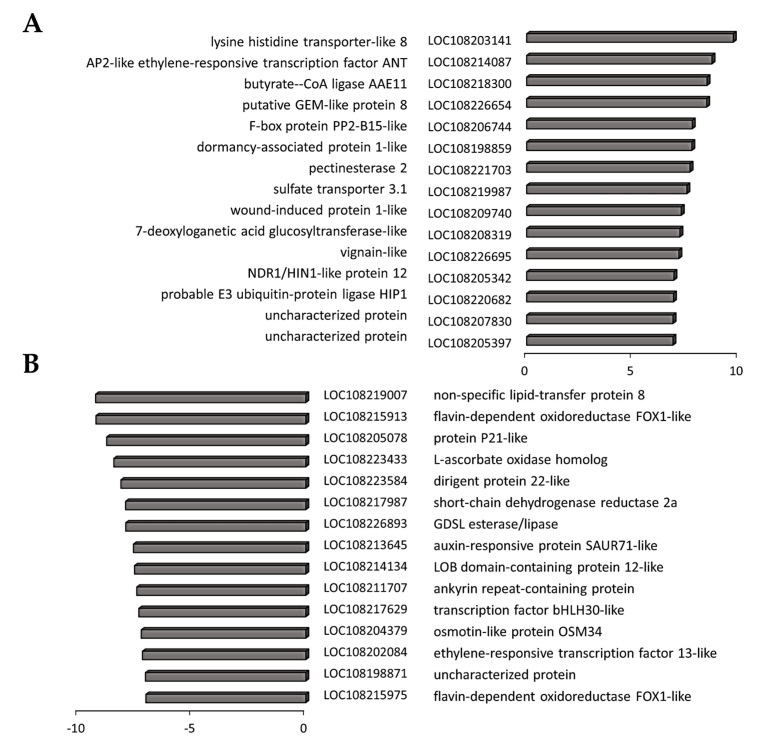
Top 15 most increased (**A**) and decreased (**B**) transcripts in black carrot tissues compared to border tissues. Three biological replicates. Three biological replicates. *p*-values were calculated using Deseq2 tool and later adjusted using the Benjamini–Hochberg procedure (*p* < 0.05; fold change ≥ 1). Values are expressed as log2(Fold change).

**Figure 10 cells-12-02465-f010:**
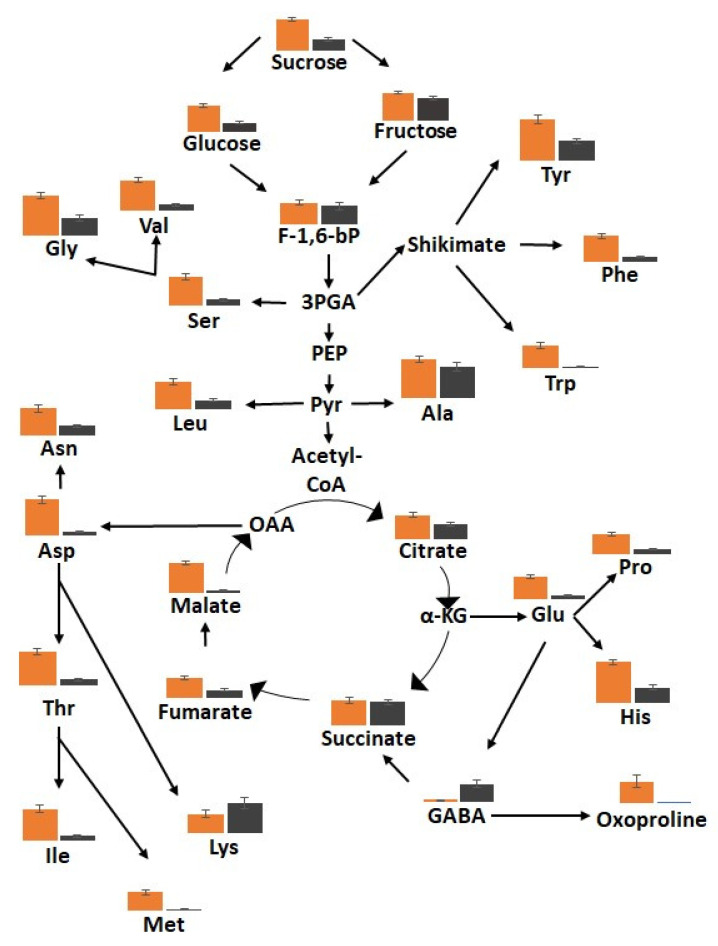
The relative levels of metabolites in the sugar, amino acid and tricarboxylic acid (TCA) pathways in the orange (orange bars) and black (black bars) tissues in the batons.

**Figure 11 cells-12-02465-f011:**
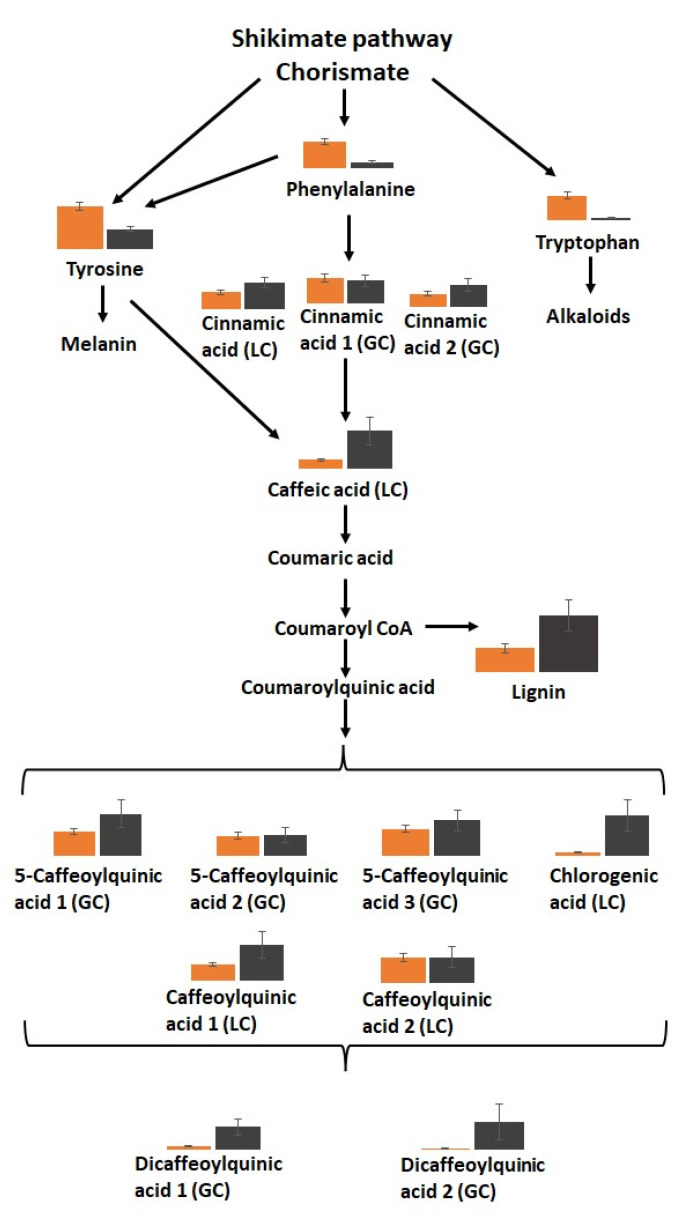
The relative levels of phenolic secondary metabolites in the orange (orange bars) pathways and black (black bars) tissues in the batons.

**Figure 12 cells-12-02465-f012:**
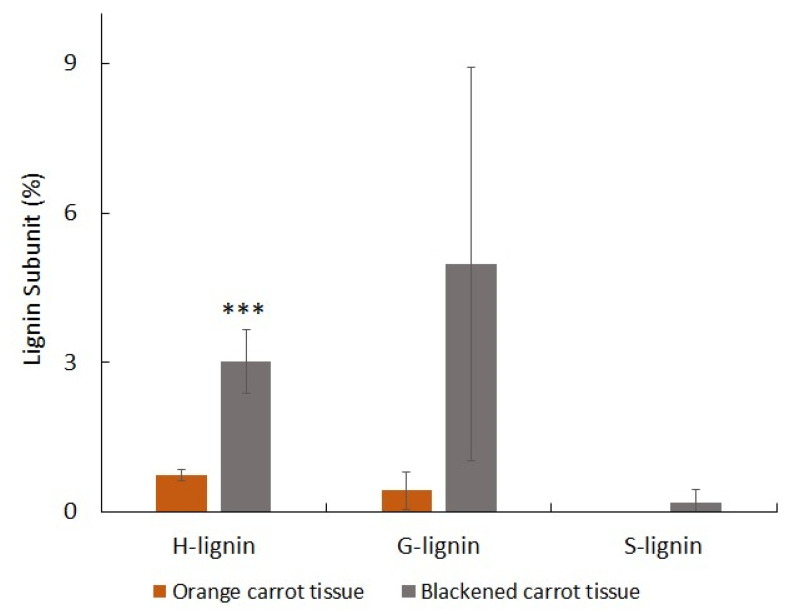
A comparison of the lignin subunit contents (%) of orange and blackened tissues of carrot batons. The lignin subunits *p*-hydroxyphenyl (H-lignin), guaiacyl (G-lignin), and syringyl (S-lignin) were determined by pyrolysis. Data are the mean values ± SD (*n* = 4). Significant differences between orange and black samples were calculated with a T-test, *** *p* < 0.001.

## Data Availability

Original data are available from the authors upon request. RNAseq data can be found in the repository Sequence Read Archive (SRA) of the National Center for Biotechnology Information (NCBI) under the project number PRJNA966197 and submission number SUB13217137.

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
