# Peer review of "Restraining Quiescence Release-Related Ageing in Plant Cells: A Case Study in Carrot"

_cells, 2023, doi:10.3390/cells12202465_

Round 1
Reviewer 1 Report (Previous Reviewer 2)
I thank the authors for accepting criticism. I believe that now the article is more logical and unambiguously proves what the authors claim.
I said that the design, requires a few corrections: Fig. 1 cd and 2, remove the dimension Bar (leave only the line) from the photos themselves, leave dimension only in the caption to the figure. Fig 3,4,5 - 10x(20) magnification - unnecessary information, as there is Bar.
And the data in Figure 6c are not clear - it is written that statistical processing was carried out, but there is no data on the graph - no reliable differences were found?
Author Response
I thank the authors for accepting criticism. I believe that now the article is more logical and unambiguously proves what the authors claim.
Authors: We thank the reviewer for these positive comments
I said that the design, requires a few corrections: Fig. 1 cd and 2, remove the dimension Bar (leave only the line) from the photos themselves, leave dimension only in the caption to the figure. Fig 3,4,5 - 10x(20) magnification - unnecessary information, as there is Bar.
Authors: We have revised the figures accordingly.
And the data in Figure 6c are not clear - it is written that statistical processing was carried out, but there is no data on the graph - no reliable differences were found?
Authors: We have removed Figure 6c because it included no new information.
Reviewer 2 Report (New Reviewer)
The introduction is rather long and does not appear to have a focus toward the research presented. The final paragraph of the introduction could be rewritten as the transition from line 102 to 103 does flow.
Line 116 is an awkward sentence.
Line 602 “obeved” ??
There are other editorial issues that need to be corrected prior to publication. The authors need to review the manuscript for errors.
There is major concern with the methods. The authors describe mapping of the RNA-seq data based on the full annotation of Machaj and Grzebelus (2021). Examination of the Machaj and Grzebelus manuscript reveals a study that examined the AHL genes in carrots and there is no mention of a genome annotation. Therefore, what did the authors map the transcriptional data to and what annotation was used to describe the genes?
The authors have written a interested paper that adds significant data to carrot post-harvest physiology. It is unclear if the blackening is a function of aging or a factor of storage conditions and perhaps the authors should note that in their manuscript. The issue regarding the reference genome used to map the transcriptional profiles should be addressed be ore publication.
The introduction is rather long and does not appear to have a focus toward the research presented. The final paragraph of the introduction could be rewritten as the transition from line 102 to 103 does flow.
Line 116 is an awkward sentence.
Line 602 “obeved” ??
There are other editorial issues that need to be corrected prior to publication. The authors need to review the manuscript for errors.
Author Response
The introduction is rather long and does not appear to have a focus toward the research presented. The final paragraph of the introduction could be rewritten as the transition from line 102 to 103 does flow.
Authors: We have revised the introduction to make the content more concise and improve the flow of information. The Introduction has evolved to address the points raised in previous reviews. We have addressed the issue with the final paragraph of the introduction, as requested.
Line 116 is an awkward sentence.
Authors: We have revised the text accordingly.
Line 602 “obeved” ??
Authors: Amended
There are other editorial issues that need to be corrected prior to publication. The authors need to review the manuscript for errors.
There is major concern with the methods. The authors describe mapping of the RNA-seq data based on the full annotation of Machaj and Grzebelus (2021). Examination of the Machaj and Grzebelus manuscript reveals a study that examined the AHL genes in carrots and there is no mention of a genome annotation. Therefore, what did the authors map the transcriptional data to and what annotation was used to describe the genes?
Authors: We apologise for the lack of clarity. RNA-Seq data was mapped to carrot reference genome ASM162521v1 ([28]; Genebank: GCF_001625215.1) - as described in this manuscript. Both the reference genome and the reference annotation of genes and transcripts was used to identify differentially expressed genes.
Gene Onthology Analysis is a separate, additional analysis, in which we used the list of differentially expressed genes. Functional annotation was generated using GO FEAD software by Machaj & Grzebelus.
These points are now clear in the revised text.
The authors have written a interested paper that adds significant data to carrot post-harvest physiology. It is unclear if the blackening is a function of aging or a factor of storage conditions and perhaps the authors should note that in their manuscript. The issue regarding the reference genome used to map the transcriptional profiles should be addressed be ore publication.
Authors: We thank the reviewer for these positive comments and we have revised the manuscript accordingly.
This manuscript is a resubmission of an earlier submission. The following is a list of the peer review reports and author responses from that submission.
Round 1
Reviewer 1 Report
The manuscript is suitable for publication in Cells.
Author Response
We thank the reviewer for this supportive comment.
Reviewer 2 Report
I started reading the authors' revised manuscript with interest, but it contains only minor revisions to the text and does not sway the essence of the experiment. The authors do not agree with all my remarks, but the article is not a field for debate, and we will leave it to the editorial board to decide. The authors write that they suggested some ways to prevent blackening of root crops. Where are the concrete experiments proving this????? Nothing but empty speculation. Even the expression of genes of phytohormones does not mean that they work, because their receptors can be inhibited, and for example ethylene autocatalysis is known, when ethylene itself induces the production of new ethylene, but it may not work due to the absence or inhibition of receptors, IAA is also capable of starting ethylene synthesis. Thus, the authors did not make any contribution to the formation of possible mechanisms to prevent carrot blackening, as well as the exact mechanisms involved in this process. Well, once again I would like to wish the authors to eat fresh fruits and vegetables, not two-year old roots, in which there is no use, which is also known, as vitamins and bioactive substances after six months of storage decrease in concentration by 30-50%, and what the authors of the work are silent about.
Author Response
We appreciate the very useful comments of this the reviewer. However, we have emphasised the key point that unlike the situation in animals there is a dearth of literature on the ageing process in plant cells. For our carrot example, clearly the simplest way to prevent blackening of the product on the supermarket the shelves is not to leave harvesting the crop for extremely long periods. More work is required to understand the plant cell ageing process and how it affects cell biology in the absence of activated developmental programs such as senescence. We speculate that the answer resides at least in part in changes in phytohormone homeostasis and cell wall metabolism. We could not find any literature showing that changes in cell wall metabolism plays a role in blackening. We hope that the reviewer will agree that these data make a new and useful contribution to the formation of possible mechanisms that underpin carrot blackening
We fully agree that society should focus more on eating fresh fruits and vegetables, and that agro-industries will modify their practices accordingly. Our hope is that the data in this manuscript will influence the thinking in food processing industries accordingly.
Reviewer 3 Report
It is interesting work, while there are still some issues need to clear. We suggest a major modification. There are some issues that need to be further discussed.
The pectin introduction should be introduced, based on the pectin structure different, please refer this reference (Food Hydrocolloids,143(2023): 108901.)
The image resolution should be improved.
Quantitative Analysis of Polyphenols in the tissue (Food Chemistry, 402(2023): 134231.).
The reference should be updated in recent years.
It is interesting work, while there are still some issues need to clear. We suggest a major modification. There are some issues that need to be further discussed.
The pectin introduction should be introduced, based on the pectin structure different, please refer this reference (Food Hydrocolloids,143(2023): 108901.)
The image resolution should be improved.
Quantitative Analysis of Polyphenols in the tissue (Food Chemistry, 402(2023): 134231.).
The reference should be updated in recent years.
Author Response
It is interesting work, while there are still some issues need to clear. We suggest a major modification. There are some issues that need to be further discussed.
Authors: We thank the reviewer for the supportive comments.
The pectin introduction should be introduced, based on the pectin structure different, please refer this reference (Food Hydrocolloids,143(2023): 108901.)
Authors: We have added this information, as requested.
The image resolution should be improved.
Authors: We have improved image quality, as requested.
Quantitative Analysis of Polyphenols in the tissue (Food Chemistry, 402(2023): 134231.).
The reference should be updated in recent years.
Authors: We have updated and improved the reference list, as requested.
We have checked the quality of English Language throughout the text.
Round 2
Reviewer 2 Report
I still cannot agree with the conclusions of the paper, in which the authors link the changes they observed to the hormonal status of root crops. The point is that hormones were not directly measured in the study, and it is not correct to talk about the effect of hormones on the basis of gene expression. There is a long chain from expression to hormone action; a hormone can be synthesized but not act, because its receptors can be inhibited. In this I see a huge drawback of the work, it turns out that the conclusions of the authors are speculative and they did not achieve the stated goals of the study. This discrepancy needs to be resolved, and the most appropriate way would be to measure hormones directly, rather than cosmetic changes in the text that do not change the value of the study itself. In addition, there are some minor design issues, but they are minor compared to the main problem of the manuscript. I do not believe that this article can be accepted for publication in a respected journal.
Author Response
We fully respect the authors comments regarding the conclusions off our paper. We fully agree that the data do not directly link the changes that were observed to the hormonal status of root crops. We have therefore changed the conclusions in line with the data that suggest that processing enhances dormancy release in the carrots. The data clearly show that changes in cell wall composition are linked to blackening and we have hence modified the conclusions accordingly. We have also thus suggested that an appropriate mechanism to avoid blackening would be to treat with nitric oxide. We trust the reviewer will now find that the conclusions are supported by the data.